# Influence-Augmented Online Planning for Complex Environments

**Jinke He**
Department of Intelligent Systems
Delft University of Technology
J.He-4@tudelft.nl

**Miguel Suau**
Department of Intelligent Systems
Delft University of Technology
M.SuaudeCastro@tudelft.nl

**Frans A. Oliehoek**
Department of Intelligent Systems
Delft University of Technology
F.A.Oliehoek@tudelft.nl

## Abstract

How can we plan efficiently in real time to control an agent in a complex environment that may involve many other agents? While existing sample-based planners have enjoyed empirical success in large POMDPs, their performance heavily relies on a fast simulator. However, real-world scenarios are complex in nature and their simulators are often computationally demanding, which severely limits the performance of online planners. In this work, we propose influence-augmented online planning, a principled method to transform a factored simulator of the entire environment into a local simulator that samples only the state variables that are most relevant to the observation and reward of the planning agent and captures the incoming influence from the rest of the environment using machine learning methods. Our main experimental results show that planning on this less accurate but much faster local simulator with POMCP leads to higher real-time planning performance than planning on the simulator that models the entire environment.

## 1 Introduction

We consider the online planning setting where we control an agent in a complex environment that is partially observable and may involve many other agents. When the policies of other agents are known, the entire environment can be modeled as a Partially Observable Markov Decision Process (POMDP) (Kaelbling et al., 1998), and traditional online planning approaches can be applied. While sample-based planners like POMCP (Silver and Veness, 2010) have been shown effective for large POMDPs, their performance relies heavily on a fast simulator to perform a vast number of Monte Carlo simulations in a step. However, many real-world scenarios are complex in nature, making simulators that capture the dynamics of the entire environment extremely computationally demanding and hence preventing existing planners from being useful in practice. Towards effective planning in realistic scenarios, this work is motivated by the question: can we significantly speed up a simulator by replacing the part of the environment that is less important with an approximate learned model?

We build on the multi-agent decision making literature that tries to identify compact representations of complex environments for an agent to make optimal decisions (Becker et al., 2003, 2004; Petrik and Zilberstein, 2009; Witwicki and Durfee, 2010). These methods exploit the fact that in many structured domains, only a small set of (state) variables, which we call *local (state) factors*, of the environment directly affects the observation and reward of the agent. The rest of the environment can only impact the agent indirectly through their influence on the local factors. For example, Figure 1a

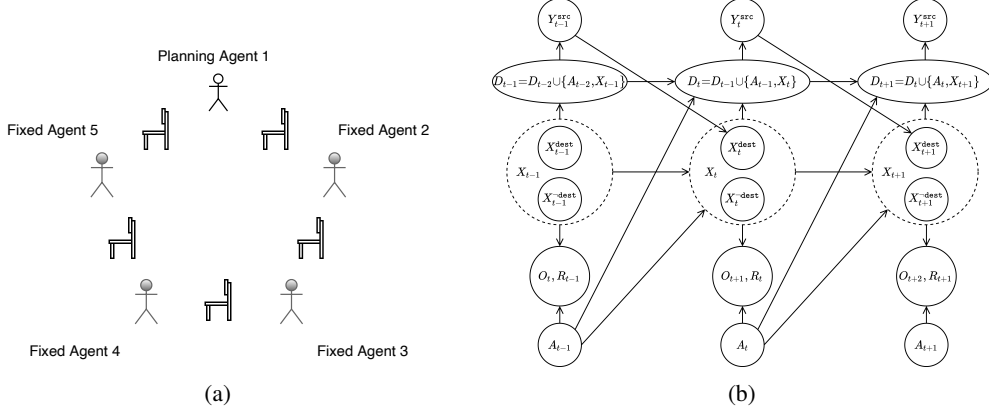

Figure 1: Left: Controlling a single agent in the Grab A Chair game with $4$ other agents. Right: Dynamic Bayesian Network for the influence-augmented local model.

shows a game called Grab A Chair, in which there are $N$ agents that, at every time step, need to decide whether they will try to grab the chair on their left or right side. An agent can only secure a chair if that chair is not targeted by the other neighboring agent. At the end of every step, each agent only observes whether it obtains the chair, without knowing the decisions of others. Additionally, there is a noise on observation, i.e., a chance that the agent gets an incorrect observation. In this game, it is clear that to the planning agent, whose goal is to obtain a chair at as many steps as possible, the decisions of neighboring agents 2 and 5 are more important than those of agents 3 and 4 as the former directly determine if the planning agent can secure a chair. In other words, only agents 2 and 5 directly *influence* agent 1's local decision making, while agents 3 and 4 may only do so indirectly.

To utilize this fact, we propose influence-augmented online planning, a principled method that transforms a factored simulator of the entire environment, called *global simulator*, into a faster *influence-augmented local simulator (IALS)*. The IALS simulates only the local factors, and concisely captures the influence of the external factors by predicting only the subset of them, called *source factors*, that directly affect the local factors. Using off-the-shelf supervised learning methods, the influence predictor is learned offline with data collected from the global simulator. Our intuition is that when planning with sample-based planners, the advantage that substantially more simulations can be performed in the IALS may outweigh the simulation inaccuracy caused by approximating the incoming influence. In this paper, we investigate this hypothesis, and show that this approach can indeed lead to improved online planning performance.

In detail, our planning experiments with POMCP show that, by replacing the global simulator with an IALS that learns the incoming influence with a recurrent neural network (RNN), we achieve matching performance while using much less time. More importantly, our real-time online planning experiments show that planning with the less accurate but much faster IALS yields better performance than planning with the global simulator in a complex environment, when the planning time per step is constrained. In addition, we find that learning an accurate influence predictor is more important for good performance when the local planning problem is tightly coupled with the rest of the environment.

## 2 Background

### 2.1 POMDP

A Partially Observable Markov Decision Process (POMDP) (Kaelbling et al., 1998) models the interactive process of an agent making decisions and receiving feedback in an environment with limited observation. Formally, a POMDP is a tuple $\mathcal{M} = (\mathcal{S}, \mathcal{A}, \mathcal{T}, \mathcal{R}, \Omega, \mathcal{O}, b_0, \gamma)$ where $\mathcal{S}$, $\mathcal{A}$, $\Omega$ are the set of environment states, actions and observations. The transition function $\mathcal{T} : \mathcal{S} \times \mathcal{A} \to \Delta(S)$ determines the distribution over the next state $S_{t+1}$ given the previous state $S_t$ and action $A_t$, where $\Delta(S)$ denotes the space of probability distributions over $S$. On transition, the agent receives a reward $R_t \sim \mathcal{R}(S_{t+1}, A_t)$ and a new observation $O_{t+1} \sim \mathcal{O}(S_{t+1}, A_t)$. A policy $\pi$ is a behavioral strategy that maps an action-observation history $h_t = \{s_0, o_1, \ldots, a_{t-1}, o_t\}$ to a distribution over actions. The

belief state $b_t \in \Delta(S)$ is a sufficient statistic of the history $h_t$, representing the distribution over $S_t$ conditioned on $h_t$, with $b_0$ being the initial belief and known. The value function $V^\pi(h_t)$ measures the expected discounted return from $h_t$ by following $\pi$ afterwards, $V^\pi(h_t) = \mathbb{E}_\pi[\sum_{k=0}^\infty \gamma^k R_{t+k} | H_t = h_t]$, where $\gamma \in [0, 1]$ is the discount factor, with the optimal value function $V^*(h_t) = \max_\pi V^\pi(h_t)$ measuring the maximally achievable value from $h_t$. The optimal value of a POMDP $\mathcal{M}$ is defined as $V_\mathcal{M}^* = V^*(b_0) = \max_\pi \mathbb{E}_\pi[\sum_{t=0}^\infty \gamma^t R_t]$.

In structured domains, the state space $\mathcal{S}$ of a POMDP can be factorized into a finite set of state variables $\mathcal{S} = \{\mathcal{S}^1, \dots, \mathcal{S}^N\}$, whose conditional independence between each other and the observation and reward variables can be utilized to construct a more compact representation of the POMDP called Dynamic Bayesian Network (DBN) (Boutilier et al., 1999). For convenience, we use the notation $S_t$ to refer to both the set of state variables and the joint random variable over them.

## 2.2 Sample-based Online Planning in POMDPs

Many real-world decision making problems are so complex that finding a policy that performs well in all situations is not possible. In such cases, online planning methods which aim to find a local policy $\pi(\cdot|h_t)$ that maximizes $V^\pi(h_t)$ when observing a history $h_t$ can lead to better performance. In fully observable case, sample-based planning methods that evaluate actions by performing sample-based lookahead in a simulator have been shown effective for large problems with sample complexity irrelevant to the state space size (Kearns et al., 2002). Monte Carlo Tree Search (MCTS) is a popular family of sample-based planning methods (Coulom, 2006; Kocsis and Szepesvári, 2006; Browne et al., 2012) that implement a highly selective search by building a lookahead tree and focusing the search on the most promising branches during the planning process.

POMCP proposed by Silver and Veness (2010) extends MCTS to large POMDPs, addressing both the curse of dimensionality and the curse of history with Monte Carlo simulation in a generative simulator $\mathcal{G}$ that samples transitions. To avoid the expensive Bayesian belief update, POMCP approximates the belief state with an unweighted particle filter. Similar to MCTS, POMCP maintains a lookahead tree with nodes representing the simulated histories $h$ that follow the real history $h_t$. To plan for an action, POMCP repeatedly samples states from the particle pool $B(h_t)$ at the root node. By simulating a state to the end, with actions selected by the UCB1 algorithm (Auer et al., 2002) inside the tree and a random policy during the rollout, the visited nodes are updated with the simulated return and the tree is expanded with the first newly encountered history. When the planning terminates, POMCP executes the action $a_t$ with the highest average return and prunes the tree by making the history $h_{t+1} = h_t a_t o_{t+1}$ the new root node. Notably, POMCP shares the simulations between tree search and belief update by maintaining a pool of encountered particles in every node during the tree search. This way, when $h_{t+1}$ is made the new root node, $B(h_{t+1})$ becomes the new estimated belief state.

## 2.3 Influence-Based Abstraction

Influence-Based Abstraction (IBA) (Oliehoek et al., 2012) is a state abstraction method (Li et al., 2006) which abstracts away state variables that do not directly affect the observation and reward of the agent, without a loss in the value. In the following, we provide a brief introduction on IBA and refer interested readers to Oliehoek et al. (2019) for more details.

Given a factored POMDP, which we call the global model $\mathcal{M}_{\texttt{global}} = (\mathcal{S}, \mathcal{A}, \mathcal{T}, \mathcal{R}, \Omega, \mathcal{O}, b_0, \gamma)$, IBA splits the set of state variables that constitute the state space $\mathcal{S}$ into two disjoint subsets, the set of *local state variables* $X$ that include at least the parent variables of the observation and reward variables and the set of *non-local state variables* $Y = \mathcal{S} \backslash X$.

IBA then defines an influence-augmented local model (IALM) $\mathcal{M}_{\texttt{IALM}}$, where the non-local state variables $Y$ are marginalized out. To define the transition function $\mathcal{T}^{\texttt{IALM}}$ on only the local state variables $X$, IBA differentiates the local state variables $X^{\texttt{dest}} \subseteq X$ that are directly affected by the non-local state variables, called *influence destination state variables*, from those that are not $X^{\neg\texttt{dest}} = X \backslash X^{\texttt{dest}}$. In addition, IBA defines the non-local state variables $Y^{\texttt{src}} \subseteq Y$ that directly affect $X$ as *influence source state variables*. In other words, the non-local state variables $Y$ influences the local state variables $X$ only through $Y^{\texttt{src}}$ affecting $X^{\texttt{dest}}$ as shown in Figure 1b. Since abstracting away $Y_0, \dots, Y_{t-1}$ creates a dependency of $Y_t^{\texttt{src}}$ on the history of local states and actions, the state $S_t^{\texttt{IALM}}$ needs to include both the local state $X_t = (X_t^{\neg\texttt{dest}}, X_t^{\texttt{dest}})$ and the so-called d-separation set

$D_t$, which encodes the relevant parts of the local history. Given this, $\mathcal{T}^{\mathtt{IALM}}$ is defined as follows:

$$\mathcal{T}^{\mathtt{IALM}}(S_{t+1}^{\mathtt{IALM}}|S_t^{\mathtt{IALM}}, A_t) = \Pr(X_{t+1}, D_{t+1}|X_t, D_t, A_t)$$
$$= \Pr(X_{t+1}^{\neg\mathtt{dest}}|X_t, A_t)\mathbb{1}(D_{t+1} = d(X_t, A_t, X_{t+1}, D_t))\Pr(X_{t+1}^{\mathtt{dest}}|X_t, D_t, A_t)$$
$$= \Pr(X_{t+1}^{\neg\mathtt{dest}}|X_t, A_t)\mathbb{1}(D_{t+1} = d(X_t, A_t, X_{t+1}, D_t))\sum_{y_t^{\mathtt{src}}} I(y_t^{\mathtt{src}}|D_t)\Pr(X_{t+1}^{\mathtt{dest}}|X_t, y_t^{\mathtt{src}}, A_t)$$

where $\mathbb{1}(\cdot)$ is the indicator function and the notation $I$ is introduced as the influence predictor, $I(y_t^{\mathtt{src}}|D_t) = \Pr(y_t^{\mathtt{src}}|D_t)$. The function $d$ selects those variables that are relevant to predict the influence sources $Y_t^{src}$. In this paper, we set $d(X_t, A_t, X_{t+1}, D_t)) = D_t \cup \{A_{t-1}, X_t\}$. That is, even though in general it is possible to condition on the history of a subset of local states and actions, we just use the entire history of local states and actions for simplicity (see Suau et al. (2020) for an exploitation of this aspect of IBA in the context of Deep RL). The IALM is then formally defined as $\mathcal{M}_{\mathtt{IALM}} = (\mathcal{S}^{\mathtt{IALM}}, \mathcal{A}, \mathcal{T}^{\mathtt{IALM}}, \mathcal{R}, \Omega, \mathcal{O}, b_0, \gamma)$ where the observation function $\mathcal{O}$, reward function $\mathcal{R}$ and the initial belief $b_0$ remain unchanged because of the definition of the local state variables $X$. Theorem 1 in Oliehoek et al. (2019) proves that this is a lossless abstraction by showing the optimal value of the IALM matches that of the global model, $V_{\mathcal{M}_{\mathtt{IALM}}}^* = V_{\mathcal{M}_{\mathtt{global}}}^*$.

## 3 Influence Augmented Online Planning

While IBA results in an IALM $\mathcal{M}_{\mathtt{IALM}}$ that abstracts away non-local state variables $Y$ in a lossless way, it is not useful in practice because computing the distribution $I(Y_t^{\mathtt{src}}|D_t)$ exactly is in general intractable. Our approach trades off between the time spent before and during the online planning, by approximating $I(Y_t^{\mathtt{src}}|D_t)$ with a function approximator $\hat{I}_\theta$ learned offline. The learned influence predictor $\hat{I}_\theta$ will then be integrated with an accurate local simulator $\mathcal{G}_{\mathtt{local}}$ to construct an influence-augmented local simulator (IALS) that only simulates the local state variables $X$ but concisely captures the influence of the non-local state variables $Y$ by predicting the influence source state variables $Y^{\mathtt{src}}$ with $\hat{I}_\theta$. During the online planning, the integrated IALS will be used to replace the accurate but slow global simulator to speed up the simulations for the sample-based online planners.

Our motivation is that by simulating the local transitions that directly decide the observation and reward of the agent with an accurate local simulator, the simulation inaccuracy caused by approximating the distribution $I(Y_t^{\mathtt{src}}|D_t)$ with $\hat{I}_\theta$ can be overcome by the advantage that simulations can be performed significantly faster in the IALS, which is essential to sample-based planners like POMCP (Silver and Veness, 2010), leading to improved online planning performance in realistic scenarios with limited planning time. Our overall approach, influence-augmented online planning, is presented in Algorithm 1, followed by our method to learn an approximate influence predictor with recurrent neural networks (RNNs) (Hochreiter and Schmidhuber, 1997; Cho et al., 2014) and integrate it with a local simulator to form a plannable IALS for sample-based planners.

### 3.1 Learning Approximate Influence Predictor Offline with RNNs

The dependency of $I(Y_t^{\mathtt{src}}|D_t)$ on the d-separation set $D_t$ renders it infeasible to be computed exactly online or offline. In this work we learn an approximate influence predictor offline with RNNs by formalizing it as a supervised sequential classification problem.

For planning with horizon $\mathcal{H}$, we need to predict the conditional distribution over the influence source state $I(Y_t^{\mathtt{src}}|D_t)$ for $t = 1$ to $\mathcal{H}-1$. We do not need to predict $I(Y_0^{\mathtt{src}}|D_0)$ as it is the initial belief over the influence source state. As RNNs require the input size to be constant for every time step, we drop the initial local state $X_0$ from $D_t$ so that the input to RNNs at time step $t$ is $\{A_{t-1}, X_t\}$ and the target is $Y_t^{\mathtt{src}}$. If there exists a distribution from which we can sample a dataset $\mathcal{D}$ of input sequence $D_{\mathcal{H}-1}$ and target sequence $(Y_1^{\mathtt{src}}, \ldots, Y_{\mathcal{H}_1}^{\mathtt{src}})$, then this is a classic sequential classification setup that can be learned by training a RNN $\hat{I}_\theta$ to minimize the average empirical KL divergence between $I(\cdot|D_t)$ and $\hat{I}_\theta(\cdot|D_t)$ with stochastic gradient descent (SGD) (Ruder, 2016), which yields a cross-entropy loss in practice. While we leave the question on how can we collect the dataset $\mathcal{D}$ in a way that maximizes the online planning performance for future investigation, in this paper we use a uniform random policy to sample $\mathcal{D}$ from the global simulator $\mathcal{G}_{\mathtt{global}}$.

---
**Algorithm 1:** Influence-Augmented Online Planning
---
**input :** a real environment `env`
**input :** a global simulator $\mathcal{G}_{\texttt{global}}$ and a local simulator $\mathcal{G}_{\texttt{local}}$
**input :** an exploratory policy $\pi_{\texttt{explore}}$
**input :** a sample-based planner `planner` with a termination condition $T$, e.g., a fixed time limit
**input :** a planning horizon $\mathcal{H}$
**Offline** *Influence Learning*

> Collect a dataset $\mathcal{D}$ of input sequences $D_{\mathcal{H}-1}=(A_{i-1}, X_i)_{i=1}^{\mathcal{H}-1}$ and target sequences
> $(Y_i^{\texttt{src}})_{i=1}^{\mathcal{H}-1}$ by interacting with the global simulator $\mathcal{G}_{\texttt{global}}$ using the policy $\pi_{\texttt{explore}}$;
> Train an approximate influence predictor $\hat{I}_\theta$ on the dataset $\mathcal{D}$ by minimizing the average
> empirical KL Divergence between $I(\cdot|D_t)$ and $\hat{I}_\theta(\cdot|D_t)$ ;

**Online** *Planning with a sample-based planner*

> Integrate the local simulator $\mathcal{G}_{\texttt{local}}$ and the learned influence predictor $\hat{I}_\theta$ into an IALS $\mathcal{G}_{\texttt{IALM}}^\theta$;
> **for** $t = 0, \ldots, \mathcal{H}-1$ **do**
>> plan for an action until $T$ is met: $a_t = \texttt{planner.plan}(\mathcal{G}_{\texttt{IALM}}^\theta, T)$;
>> execute the action in the real environment: $o_{t+1} = \texttt{env.act}(a_t)$ ;
>> process the new observation: $\texttt{planner.observe}(o_{t+1})$
> **end**
---

## 3.2 Integrating the Local Simulator and RNN Influence Predictor for Online Planning

To plan online in a POMDP, sample-based planners like POMCP (Silver and Veness, 2010) require a generative simulator that supports sampling the initial states and transitions. As shown in Figure 1b, to sample a transition in the IALS $\mathcal{G}_{\texttt{IALM}}^\theta$, we need to first sample an influence source state $Y_t^{\texttt{src}}$ and then sample the local transitions in the local simulator $\mathcal{G}_{\texttt{local}}$. While in the original formulation of IBA, $\hat{I}_\theta(Y_t^{\texttt{src}}|D_t)$ conditions on the d-separation set $D_t$ which grows with actions $A_t$ and new local states $X_{t+1}$ at every time step, we avoid feeding the entire $D_t$ into RNNs for every prediction of $Y_t^{\texttt{src}}$ by taking the advantage of RNNs whose hidden state $Z_t$ is a sufficient statistic of the previous inputs. As a result, we use $S_t^{\texttt{IALM}} = (X_t, Y_t^{\texttt{src}}, Z_t)$ as the state of the IALS in practice. The transition $s_{t+1}^{\texttt{IALM}}, o_{t+1}, r_{t+1} \sim \mathcal{G}_{\texttt{IALM}}^\theta(s_t^{\texttt{IALM}}, a_t)$ can then be sampled in two steps:

- sample the next local state, observation and reward: $x_{t+1}, o_{t+1}, r_t \sim \mathcal{G}_{\texttt{local}}(x_t, y_t^{\texttt{src}}, a_t)$
- sample the next RNN hidden state and influence source state : $z_{t+1}, y_{t+1}^{\texttt{src}} \sim \hat{I}_\theta(\cdot|z_t, a_t, x_{t+1})$

The initial state $S_0^{\texttt{IALM}}$ of the IALS can be easily sampled by first sampling a full state $s \sim b_0$ and then extracting the local state and the influence source state $(x_0, y_0^{\texttt{src}})$ from $s$.

# 4 Experiments

We perform online planning experiments with the POMCP planner (Silver and Veness, 2010) to answer the following questions: when learning approximate influence predictors with RNNs,

> - can planning with an IALS be faster than planning with the global simulator while achieving similar performance, *when the same number of simulations are allowed per planning step*?
> - can planning with an IALS yield better performance than planning with the global simulator, *when the same amount of planning time is allowed per planning step*?

**Experimental Setup**

Our codebase was implemented in C++, including a POMCP planner and several benchmarking domains [1]. We ran each of our experiments for many times on a computer cluster with the same amount of computational resources. To report results, we plot the means of evaluation metrics with standard errors as error bars. Details of our experiments are provided in the supplementary material.

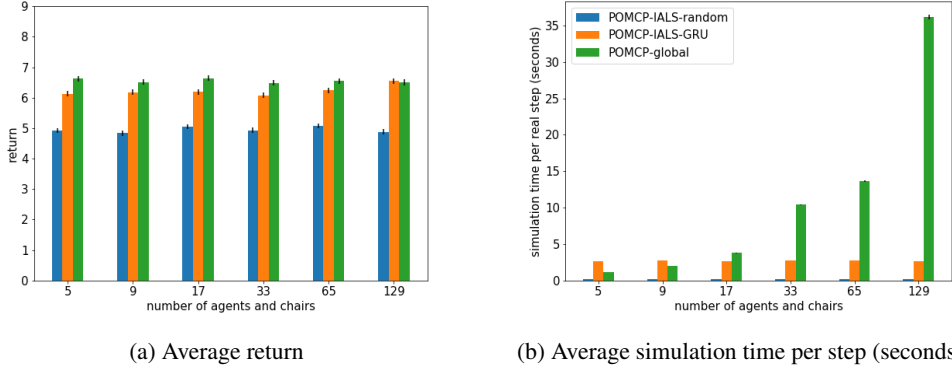

(a) Average return

(b) Average simulation time per step (seconds)

Figure 2: Performance of POMCP with different simulators in Grab A Chair games of various sizes. While the IALS with GRU influence predictor achieves matching returns with the global simulator, the simulation is significantly faster in scenarios with many other agents.

### Grab A Chair

The first domain we use is the Grab A Chair domain mentioned in Section 1. In our setting, the other agents employ a policy that selects chairs randomly in the beginning and greedily afterwards according to the frequency of observing to obtain a chair when visiting it.

Our intuition is that the amount of speedup we can achieve by replacing $\mathcal{G}_{\texttt{global}}$ with $\mathcal{G}_{\texttt{IALM}}^{\theta}$ depends on how fast we can sample influence source state variables $Y^{\texttt{src}}$ from the approximate influence predictor $\hat{I}_{\theta}$ and the size of hidden state variables $Y \backslash Y^{\texttt{src}}$ we can avoid simulating in $\mathcal{G}_{\texttt{IALM}}^{\theta}$. We perform planning with different simulators in games of $\{5, 9, 17, 33, 65, 129\}$ agents for a horizon of 10 steps, where a fixed number of 1000 Monte Carlo simulations are performed per step.

To obtain an approximate influence predictor $\hat{I}_{\theta}$, we sample a dataset $\mathcal{D}$ of 1000 episodes from the global simulator $\mathcal{G}_{\texttt{global}}$ with a uniform random policy and train a variant of RNN called Gated Recurrent Units (GRU) (Cho et al., 2014) on $\mathcal{D}$ until convergence. To test if capturing the incoming influence is essential for achieving good performance when planning on $\mathcal{G}_{\texttt{IALM}}^{\theta}$, we use an IALS with a uniform random influence predictor as an additional baseline, denoted as $\mathcal{G}_{\texttt{IALM}}^{\texttt{random}}$.

Figure 2a shows the performance of planning with different simulators in scenarios of various sizes. It is clear that planning on $\mathcal{G}_{\texttt{IALM}}^{\theta}$ achieves significantly better performance than planning on $\mathcal{G}_{\texttt{IALM}}^{\texttt{random}}$, emphasizing the importance of learning $\hat{I}_{\theta}$ to capture the influence. While planning on $\mathcal{G}_{\texttt{IALM}}^{\theta}$ can indeed achieve matching performance with $\mathcal{G}_{\texttt{global}}$ as shown by the small differences in their returns, the advantage of the IALS, its speed, is shown in Figure 2b. In contrast to $\mathcal{G}_{\texttt{global}}$ which slows down quickly because of the growing number of state variables to simulate, the computation time of both $\mathcal{G}_{\texttt{IALM}}^{\theta}$ and $\mathcal{G}_{\texttt{IALM}}^{\texttt{randomly}}$ barely increases. This is because those state variables added by more chairs and agents are abstracted away from the simulations in the IALS with their influence concisely captured by $\hat{I}_{\theta}$ in the distribution of the two neighboring agents' decisions. Note that $\mathcal{G}_{\texttt{IALM}}^{\theta}$ is slower than $\mathcal{G}_{\texttt{global}}$ in scenarios with few agents due to the overheads of feedforward passing in the GRU.

To further investigate how will influence-augmented online planning perform in environments with different *influence strengths*, by which we mean the degree to which the local states are affected by the influence source states, we repeat our experiments above in a variant of the 5-agent Grab A Chair game where the only difference is that when two agents target the same chair, both of them have the same probability $p \in [0, 1]$ to obtain the chair [2]. The intuition is that when $p$ is lower, the influence from the rest of the environment will be stronger as the decisions of the two neighboring agents will be more decisive on whether the planning agent can secure a chair. In this case, higher prediction accuracy on the decisions of the two neighboring agents will be required for the agent to plan a good action. Figure 3 shows the planning performance with all simulators under decreasing $p$ which

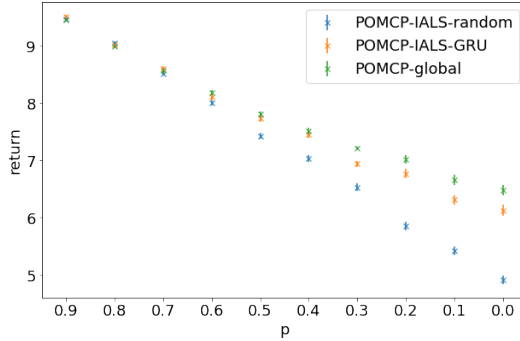

Figure 3: Performance of POMCP with different simulators in the modified Grab A Chair game under decreasing $p$, which implies stronger influence from the rest of the environment. The smaller performance difference between $\mathcal{G}_{\texttt{IALM}}^{\theta}$ and $\mathcal{G}_{\texttt{global}}$ under higher $p$ suggests that learning an accurate influence predictor is more important to achieve good planning performance when the local planning problem is more tightly coupled with the rest of the environment.

implies stronger influence strength from the rest of the environment. While the same amount of effort was put into training the approximate influence predictor $\hat{I}_{\theta}$, the performance difference between planning with $\mathcal{G}_{\texttt{IALM}}^{\theta}$ and $\mathcal{G}_{\texttt{global}}$ is smaller under higher $p$. This suggests that in environments where the local planning problem is more tightly coupled with the rest of the environment, learning an accurate influence predictor $\hat{I}_{\theta}$ is more important to achieve good planning performance.

**Real-Time Online Planning in Grid Traffic Control**

The primary motivation of our approach is to improve online planning performance in realistic settings where the planning time per step is constrained. For this reason, we conduct real-time planning experiments in a more realistic domain called Grid Traffic Control, which simulates a busy traffic system with 9 intersections, each of which consists of 4 lanes with 6 grids as shown in Figure 4a, with more details provided in the supplementary material.

The traffic lights are equipped with sensors providing 4-bit information indicating if there are vehicles in the grids around them. While the other traffic lights employ a hand-coded switching strategy that prioritizes lanes with vehicles before the lights and without vehicles after the lights, the traffic light in the center is controlled by planning, with the goal to minimize the total number of vehicles in this intersection for a horizon of 30 steps.

As mentioned in Section 2.2, POMCP approximates the belief update with an unweighted particle filter that reuses the simulations performed during the tree search. However, in our preliminary experiments, we observed the particle depletion problem, which occurred when POMCP ran out of particles because none of the existing particles was evidenced by the new observation. While to alleviate this problem we use a workaround inspired by Silver and Veness (2010) [3], when particle depletion still occurs at some point during an episode, the agent employs a uniform random policy.

We train an influence predictor with a RNN and evaluate the performance of all three simulators $\mathcal{G}_{\texttt{IALM}}^{\texttt{random}}$, $\mathcal{G}_{\texttt{IALM}}^{\theta}$ and $\mathcal{G}_{\texttt{global}}$ in settings where the allowed planning time is fixed per step. Our hypothesis is that $\mathcal{G}_{\texttt{IALM}}^{\theta}$ will outperform the $\mathcal{G}_{\texttt{global}}$ when the planning time allowed is very constrained because in that case, the advantage on simulation speed will dominate the disadvantage on simulation accuracy caused by approximating the influence with $\hat{I}_{\theta}$.

Figure 4b demonstrates the ability of the IALS to perform more than twice the number of simulations that can be performed by the global simulator within the same fixed time. This is directly translated into the ability of POMCP to plan for more time steps before the particle depletion occurs as shown in Figure 4c. The more important effect of faster simulation is that our approach performs much better

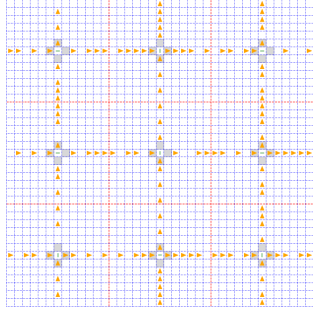

(a) The Grid Traffic Control domain

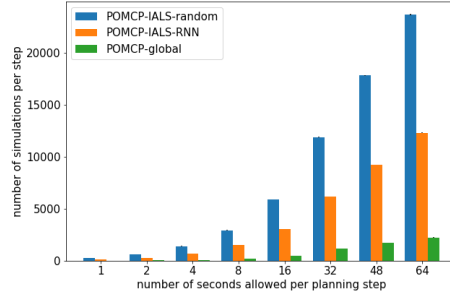

(b) Number of simulations performed per planning step

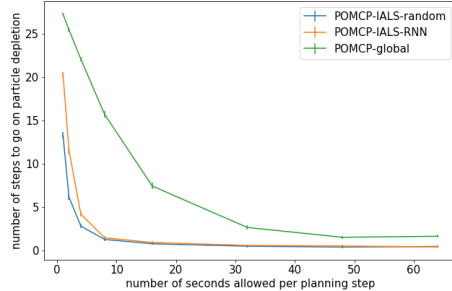

(c) Number of steps to go on particle depletion

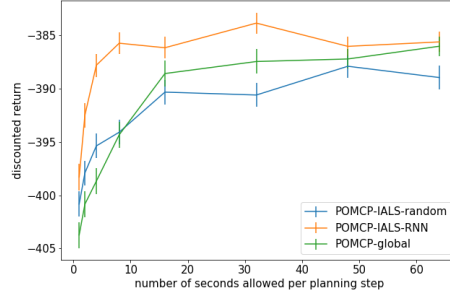

(d) Discounted return

Figure 4: Performance of POMCP with different simulators while allowing different numbers of seconds per planning step in the Grid Traffic Control domain. While the planning performance of the IALS with trained influence predictor dominates the global simulator when the planning time is constrained, the performance difference decreases when more time is allowed.

than planning on the global simulator especially when the planning time is limited. This suggests that there does exist a trade-off between simulation speed and simulation accuracy that allows planning on the IALS with an approximate influence predictor to achieve better online performance.

Figure 6 in the supplementary material performs a similar time-constrained evaluation in the Grab A Chair domain. The finding there is that the advantage of the IALS on the simulation speed is clearer when the global model of the problem is more complex, in which cases the IALS with an approximate influence predictor shows a superior performance compared to the global simulator.

## 5 Related Work

The idea of utilizing offline knowledge learning for improved online planning performance has been well-studied (Gelly and Silver, 2007, 2011; Silver et al., 2016, 2017, 2018; Anthony et al., 2017). These approaches can be categorized as 1) learning value functions or policies to guide the tree search, 2) improving default policy for more informative rollouts, 3) replacing rollouts with learned value functions and 4) initializing state-action value estimates. Our approach takes a distinct path by speeding up computationally expensive forward simulations, which allows the planner to sample more trajectories for each decision.

Closest to our work is the approach by Chitnis and Lozano-Pérez (2020), which exploits *exogenous variables* to reduce the state space of the model for more efficient simulation and planning. While both of the approaches learn a more compact model by abstracting away state variables, exogenous variables are fundamentally different from the non-local variables that we abstract away. By definition, exogenous variables refer to those variables that are beyond the control of the agent: they cannot be affected, directly or indirectly, by the agent's actions (Boutilier et al., 1999; Chitnis and Lozano-Pérez, 2020). In contrast, the non-local variables that are abstracted away in IBA (Oliehoek et al., 2012) can be chosen more freely, as long as they do not *directly* affect the agent's observation and reward. Therefore, the exogenous variables and non-local variables are in general two different sets

of variables that can be exploited to reduce the state space size. For instance, in the traffic problem of Figure 4a, there are no exogenous variables as our action can directly or indirectly effect the transitions at other intersections (by taking or sending vehicles from/to them). This demonstrates that our approach allows us to reduce the state space of this problem beyond the exogenous variables.

The idea of replacing a computationally demanding simulator with an approximate simulator for higher simulation efficiency has been explored in many fields under the name of *surrogate model*, such as computer animation (Grzeszczuk et al., 1999), network simulation (Kazer et al., 2018), the simulation of seismic waves (Moseley et al., 2018) and so on. Our work explores this idea in the context of sample-based planning in structured domains.

Recent works in deep model-based reinforcement learning (Oh et al., 2017; Farquhar et al., 2018; Hafner et al., 2019; Schrittwieser et al., 2019; Van der Pol et al., 2020) have proposed to learn an approximate model of the environment by interacting with it, and then plan a policy within the learned model for better sample efficiency. Our method considers a very different setting, in which we speed up the simulation for sample-based planning by approximating part of the global simulator, that is, the influence from the rest of the environment, and retain the simulation accuracy by explicitly utilizing a light and accurate local simulator.

# 6   Conclusion

In this work we aim to address the problem that simulators modeling the entire environment is often slow and hence not suitable for sample-based planning methods which require a vast number of Monte Carlo simulations to plan a good action. Our approach transforms an expensive factored global simulator into an influence-augmented local simulator (IALS) that is less accurate but much faster. The IALS utilizes a local simulator which accurately models the state variables that are most important to the planning agent and captures the influence from the rest of the environment with an approximate influence predictor learned offline. Our empirical results in the Grid Traffic Control domain show that in despite of the simulation inaccuracy caused by approximating the incoming influence with a recurrent neural network, planning on the IALS yields better online performance than planning on the global simulator due to the higher simulation efficiency, especially when the planning time per step is limited. While in this work we collect data from the global simulator with a random exploratory policy to learn the influence, a direction for future work is to study how this offline learning procedure can be improved for better performance during online planning.

## Broader Impact

The potential impact of this work is precisely its motivation: making online planning more useful in real-world decision making scenarios, enabling more daily decisions to be made autonomously and intelligently, with promising applications including autonomous warehouse and traffic light control.

Unlike simulators constructed by domain experts, which are in general easier to test and debug, influence-augmented local simulator contains an approximate influence predictor learned from data, which may fail with rare inputs and result in catastrophic consequences especially when controlling critical systems. This suggests that extensive testing and regulation will be required before deploying influence-augmented local simulators in real-world decision making scenarios.

## Acknowledgments and Disclosure of Funding

This project had received funding from the European Research Council (ERC) under the European Union's Horizon 2020 research and innovation programme (grant agreement No. 758824 —INFLUENCE). 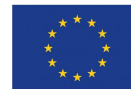 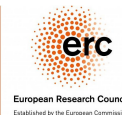

## Footnotes

[1] available at `https://github.com/INFLUENCEorg/IAOP`

[2]Note that this leads to a physically unrealistic setting since it is possible that two agents obtain the same chair at a time step. However, it gives us a way to investigate the impact of the influence strength from the rest of the environment.

[3]While more advanced particle filters like Sequential Importance Resampling can reduce this problem, we chose to use POMCP in unmodified form to make it easier to interpret the benefits of our approach. Our workaround is that when the search tree is pruned because of a new observation, we add $N/6$ additional particles sampled from the initial belief $b_0$ to the current particle pool where $N$ is the number of remaining particles.

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
