[Supplementary Material]

# 7    Supplementary Material

In this supplementary material, we provide the details of our experimental setups for reproducibility.

## 7.1    Grab A Chair

### Environment

The Grab A Chair game is an $N$-agent game where at every time step, each agent has an action space of two, trying to grab the chair on its left or right side. An agent only secures a chair if its targeted chair is not targeted by a neighboring agent. At the end of a time step $t$, each agent with $s_{t+1} \in \{0, 1\}$ indicating if this agent obtains a chair receives a reward $r_t = s_{t+1}$ and a noisy observation $o_{t+1}$ on $s_{t+1}$ which has a probability 0.2 to be flipped.

In the experiments of Figure 3, when two agents target the same chair, both of them have a probability of $p \in [0, 1]$ to secure the chair, which means that there is a probability that two neighboring agents obtain the same chair. The following setup applies to all the experiments in this domain.

### Experimental Setup

#### Influence Learning

In this domain, the approximate influence predictor $\hat{I}_\theta$ is parameterized by a GRU (Cho et al., 2014) classifier with 8 hidden units. The dataset $\mathcal{D}$ consists of 1000 episodes collected from the global simulator $\mathcal{G}_{\texttt{global}}$ with a uniform random policy, where 800 episodes are used as the training set and the other 200 episodes are used as the validation set. The hyperparameters used to train the GRU influence predictors in scenarios with $\{5, 9, 17, 33, 65, 129\}$ agents are shown in Table 1 and their learning curves are shown in Figure 5.

Table 1: Hyperparameters used to train the GRU influence predictors for experiments in the Grab A Chair domain, where the weight decay was fine tuned within the range until there is no clear sign of overfitting.

| | |
|---|---|
| Learning rate | 0.0005 |
| Batch size | 128 |
| Number of epochs | 8000 |
| Weight decay | $[1 \times 10^{-5}, 5 \times 10^{-5}]$ |

(a) 5 agents          (b) 9 agents          (c) 17 agents

(d) 33 agents          (e) 65 agents          (f) 129 agents

Figure 5: Learning curves of influence predictors in the Grab A Chair domain.

Figure 6: Performance of POMCP with different simulators while allowing different numbers of seconds per planning step in Grab A Chair games with 33 (left), 65 (middle) and 129 (right) agents. The advantage of IALS with GRU influence predictor over global simulator becomes clearer as the global model of the planning problem gets more complex.

**Planning with POMCP**

The parameters used in the planning experiments with POMCP are shown in Table 2.

Table 2: Parameters for the planning experiments with POMCP in the Grab A Chair domain.

| | |
|---|---|
| Discount factor | 1.0 |
| Horizon | 10 |
| Number of simulations per step | 1000 |
| Number of initial particles | 1000 |
| Exploration constant in the UCB1 algorithm ($c$) | 100.0 |

**Real-time Online Planning in Grab A Chair domain**

We conduct a time-constrained evaluation in this domain with $\{33, 65, 129\}$ agents, similar to the one performed in the Grid Traffic Control domain, where different amount of time is allowed to plan an action. Results in Figure 6 show that the advantage of the IALS with GRU influence predictor is clearer when the global model of the planning gets more complex.

## 7.2 Grid Traffic Control

**Environment**

The Grid Traffic Control environment simulates a traffic system of 9 intersections as shown in Figure 4a. The vehicles, plotted as yellow arrows, move from the left to right and the bottom to top, governed by the traffic lights in the center of each intersection. While they are initially generated with a probability of 0.7 in each grid, new vehicles will enter the traffic system at entrances on the left and bottom borders whenever they are not occupied at last time step. When reaching the right and bottom borders, with a probability of 0.3, vehicles leave the traffic system.

While the other traffic lights are controlled by fixed switching strategies, the traffic light in the center intersection is controlled by the planning agent, whose action space consists of actions to set the light green for each lane. After an action $a_t$ is taken which results in the movement of vehicles, the agent receives an observation consisting of four Boolean variables $o_{t+1} = \{\texttt{left\_occupied}, \texttt{right\_occupied}, \texttt{up\_occupied}, \texttt{bottom\_occupied}\}$ indicating if the four grids around the traffic light are occupied. The reward $r_t$ is the negative number of grids that are occupied in this intersection after the transition at time step $t$.

Figure 7: The learning curve of the influence predictor in the Grid Traffic Control domain.

## Experimental Setup

### Influence Learning

In this domain, the approximate influence predictor $\hat{I}_\theta$ is parameterized by a RNN classifier with 2 hidden units. The dataset $\mathcal{D}$ consists of 1000 episodes collected from the global simulator $\mathcal{G}_{\texttt{global}}$ with a uniform random policy, where 800 episodes are used as the training set and the other 200 episodes are used as the validation set. The hyperparameters used to train the RNN influence predictor are shown in Table 3 and its learning curve is shown in Figure 7.

Table 3: Hyperparameters used to train the RNN influence predictor for experiments in the Grid Traffic Control domain.

| | |
|---|---|
| Learning rate | 0.00025 |
| Batch size | 128 |
| Number of epochs | 8000 |
| Weight decay | $1 \times 10^{-4}$ |

### Planning with POMCP

The parameters used in the planning experiments with POMCP are shown in Table 4, where effective horizon is the maximal depth from the root node that a search or a rollout will be performed.

Table 4: Parameters for the planning experiments with POMCP in the Grid Traffic Control domain.

| | |
|---|---|
| Discount factor | 0.95 |
| Horizon | 30 |
| Number of seconds allowed per planning step | $\{1, 2, 4, 8, 16, 32, 48, 64\}$ |
| Number of initial particles | 1000 |
| Exploration constant in the UCB1 algorithm ($c$) | 10.0 |
| Effective horizon | 18 |