[Reviews · NeurIPS 2020]

Review 1

Summary and Contributions: New comment: I confirm that I have read all reviews and the authors' response. After pondering these and discussing with other reviewers, I maintain the current mark. The work presents an approach to online planning that simulates only part of a full state, for a faster simulation. The approach combines ideas from the literature, such as Influence-Based Abstraction (IBA), Dynamic Bayesian Networks and sample-based online planning. For example, IBA allows to split the set of state variables into so-called local variables (e.g., variables that impact the observations in a state, and variables that impact the reward) and non-local state variables. Local state variables are former partitioned into variables influenced by non-local variables, and the rest of local variables. During online planning, rather than simulating full states, IBA-specific equations (that tell how to factor the probabilisitc transition function) allow to use an approximate (localized) influence predictor, which estimates relevant non-local variables (i.e., non-local variables that directly influence local variables) based on a history of actions and observations. The simulator is implemented as a recurring neural network. Experiments are presented in two domains: a modelling of a grab-a-chair game, and controlling an intersection on a grid-like topology of roads, with nearby intersections controlled with a static policy. In the former domain, the cost of the global state simulation grows fast, slowing down a planner version based on global simulation. In the latter domain, the differences in terms of computational time between a global simulation and a local simulation are smaller and more stable, revolving around a factor of 2. Depending on the threshold for the planning time per move is set, this may or may not make a positive difference in the performance of the planner.

Strengths: The paper is clearly written and fairly easy to follow. The methodology looks clean and suitable for the purpose. The results show some benefit of local state simulation.

Weaknesses: The experimental evaluation could be more thorough. The benefits in the traffic application are rather moderate. The grab-a-chair domain seems to have some clear regular structure in a way, which makes it an interesting domain but still particular in that respect. Additional domains would add value to the evaluation.

Correctness: The claims and the method seem to be correct. Please see a comment about the empirical methodology.

Clarity: Yes, the clarity of the paper is at a good level.

Relation to Prior Work: As far as I can tell there is a reasonable discussion of prior work (but I do not have up-to-date expert knowledge in this problem).

Reproducibility: Yes

Additional Feedback:


Review 2

Summary and Contributions: This paper proposes a method that replaces a global simulator of the environment by an influence-augmented local simulator (IALTS). It is based on the premise that in many complex factored environments, such as multi-agent environments, only a small set of local factors (i.e. a small set of state variables) directly influence the policy of a planning agent, and therefore only local factors need to be simulated exactly, whereas the influence of external factors can be simulated approximately using a learned model. Experiments show that the IALTS can be used to significantly speed up an online POMDP solver (POMCP in this case) compared to when the full global simulator is used.

Strengths: I like the idea of separating the global simulator into an accurate one for the local state variables and an approximate one (using RNNs) to simulate the indirect factors. From a technical standpoint this makes a lot of sense for problems that admit a factored representation of the state space, and the experimental results show quite impressive gains achieved of the proposed method compared to just using the global simulator for the entire problem.

Weaknesses: The set of experiments is rather limited. However, I believe that the method can be applied to a larger variety of problems that admit a factored representation of the state space and where sampling states and observations from a global simulator is expensive.

Correctness: The claims and methods are correct.

Clarity: The paper is well written.

Relation to Prior Work: The paper is well situated in the literature.

Reproducibility: Yes

Additional Feedback: The submission does not provide codes that generate the results. It will likely take substantial effort to reproduce. However, the parameters used in testing are available in the supplementary materials. Regarding particle depletion problem, in many cases the particle depletion problem can be easily avoided by using a more sophisticated particle filter for the belief update (e.g., Sequential Importance Resampling). I have read the rebuttal. Thank you for the clarification. It does not change my position.


Review 3

Summary and Contributions: This work presents an approach towards POMDP planning in which only a subset of the environment has direct influence on the agent. IALM learns an influence predictor and integrates it into a local model that simulates specifically local transitions. This allows a fast simulation that is accurate for the factors that directly matter to the agent. The method is integrated as the simulator in a POMCP framework. It is demonstrated to outperform global methods when planning time is constrained for two representative problems.

Strengths: The paper is clear and the motivation is good and the method is novel. By focusing only on important parts of the simulation the method gets large speeds and loses little of what matters. The fact that it loses little of what matters is clear from Fig. 2. And that it outperforms global methods when time is limited is clear from Fig. 3.

Weaknesses: The experiments clearly demonstrate the desired behavior of the algorithm. However, they are very clear cut cases where this approach will help, where the direct and indirect factors are easy to discover and have clear separations. This is nice for exposition, but it would be good to understand the limits of this approach. How does it perform when the factors are less clear, are there substantial losses? The authors should provide more experiments really testing this boundary on a less “designed” problem. This would increase the significance of the approach substantially. It’s not clear from the paper how the supervision for Influence training supervision is computed. === After rebuttal === I appreciate the author's response. After reading the rebuttal I still believe more exhaustive evaluations would benefit.

Correctness: The method, claims, and methodology look correct.

Clarity: The paper is clear and well written.

Relation to Prior Work: The paper seems to compare reasonably with related work. There is some work in learning models that are value-, goal-, and task-aware that is related and showing promise: http://proceedings.mlr.press/v54/farahmand17a/farahmand17a.pdf, https://arxiv.org/pdf/1909.04115.pdf, https://arxiv.org/pdf/2007.07170.pdf

Reproducibility: No

Additional Feedback: The chair task should also be evaluated in a time constrained setting. It may be possible to compute the influence distribution from self-supervised data rather than supervision through an information bottleneck.


Review 4

Summary and Contributions: This paper introduced a method called influence-augmented online planning that enables efficient online probabilistic planning under partial observability. In particular, the developed approach augments the subset of factored state-space variables that have direct effect in the agent's observations and rewards. This subset extraction is done using a local simulator (influence-augmented local simulator, or IALS). The less relevant variables from the rest of the environment may still influence the variables in the local simulator. This influence is estimated by an RNN that is trained offline using the data from the global simulator. The approach was evaluated using two testing domains: grab a chair, where planning agents and other agents compete on grabbing a chair; and a traffic light domain. The experimental setup shows that the developed approach was able to compute plans in a much shorter time especially when the state space is bigger than an online POMCP planner with global simulator that simulates the whole state-space including both relevant and less relevant states. The faster simulation yields a slight drop in the accuracy of the computed plan. ------- after rebuttal ------- Thanks for the response in detail. The overall score has been increased.

Strengths: The paper is overall well written, and the theoretical grounding is strong. The scalability of the developed approach was well evaluated. There is novelty from the application perspective.

Weaknesses: The major concern is that the idea of exploiting "influences" of domain variables to reduce the state space of POMDPs is not new. In the literature, those variables that only indirectly influence agent behaviors are referred to as exogenous variables. The following are two papers that studied this idea. The RNN-based influence learning is new within the literature, while the following two papers have studied other reasoning and learning methods to incorporate exogenous variables into POMDP-based action selection processes. Zhang S, Khandelwal P, Stone P. Dynamically constructed (PO) MDPs for adaptive robot planning. AAAI 2017 Chitnis R, Lozano-Pérez T. Learning Compact Models for Planning with Exogenous Processes. CoRL 2019 The effort of data collection and annotation for offline training the RNN model should be discussed and evaluated. Less accurate influence models can potentially affect the agent's behaviors in a negative way, and good accuracy requires considerable training data. In comparison to traditional POMDP algorithms, this data collection process brings extra computation and data collection burden, and reduces its applicability. The current experimental setup consists of two baselines. One global simulator and one with uniform random influence. It's unclear if an additional baseline of local simulators with “no influence” would perform more accurately and faster than the proposed method. Grab-A-Chair problem is explained in Introduction, Experiment and the supplementary material. However, it's still unclear if it's an episodic process, and (if so) what leads to the end of an episode. In the same problem, it's unclear if the agent can grab more than 2 chairs (looks like the case). As a result, Figure 2(a) is confusing. Why don't we see increase in the return value when the number of agents and chairs increase? A couple of minor suggestions beyond this paper: Maybe the authors can investigate enabling the RNN model with attention mechanism for better prediction of the influence. Some justification on the discount factor (being 1.0) in the first experiment will be appreciated. In the 2nd experiment, the particle depletion problem was observed and N/6 particles were added to alleviate the issue. Maybe Algorithm 1 can be revised accordingly. Minor: Line 230, G_IALM → G^{random}_IALM How was hyperparameter sweeping done?

Correctness: Yes

Clarity: Adequate

Relation to Prior Work: No -- see above comments.

Reproducibility: Yes

Additional Feedback:

[Author Response · NeurIPS 2020]



Figure 1: Time-constrained evaluation in the grab a chair domain with 33 (left), 65 (middle) and 129 (right) agents.

We thank all reviewers for the encouraging and constructive feedback and the relevant pointers.

**Reproducibility** (@**R2**, **R3**): Our codebase is available at `https://www.dropbox.com/sh/sg2qyfpdwfmkfqd/`
`AACq5R5BQS-jpSS6OMFS3FrLa?dl=0` with source code, pretrained models and instructions to reproduce every figure
in the submission. We will make the codebase available on Github after the review phase.

**Comparison to exogenous variables** (@**R4**): Thanks for the pointers to the related work which indeed are similar in
spirit. We will add the discussion of these works to our related work section. Nevertheless, we would like to point out
that exogenous variables are fundamentally different from the non-local variables: by definition, exogenous variables
refer to those variables that are **beyond the control of the agent** in the sense that **the values of which are not affected,**
**directly or indirectly, by the agent's actions** (Boutilier et al., 1999; Zhang et al., 2017; Chitnis et al., 2019). However,
the non-local variables in the influence-based abstraction (IBA) (Oliehoek et al., 2012) refer to those variables that **do**
**not directly affect the agent's observation and reward**. Therefore, the exogenous variables and non-local variables
are in general two different sets of variables that can be exploited to reduce the state space size. For instance, in
the traffic problem of Figure 4a, there are *no* exogenous variables as our action can directly or indirectly effect the
transitions at other intersections (by taking or sending vehicles from/to them). In other words, **our approach allows us**
**to reduce the state space of this problem** *beyond* **the exogenous variables**.

**Clarifications on data collection and influence learning** (@**R3**, **R4**): In Section 3.1, we formalize influence prediction
as a classification problem where inputs are trajectories of local states and actions, and outputs are influence source
variables. In this work, the training trajectories are sampled **from the global simulator** (cf. line 162) with an exploratory
policy and therefore no human annotation is required.

In all our experiments, we collected 1000 episodes (line 199) and trained the influence predictor to convergence, which
did not take more than an hour. While such offline computation could be costly (as noted by **R4**), we believe that in
many settings the real-time constraints during the online phase are the bottleneck for application and these can be
significantly brought down with our approach.

**Clarifications on experimental design and results** (@**R1**, **R2**, **R3**, **R4**): As **R1** and **R3** noted, we only investigate
the performance of our approach in those domains for which it was designed: problems with clear factorized structure.
For these problems, we think our benchmarking domains are general enough so that the results can be easily translated.
We certainly would not want to claim anything about a broader class of problems. We do stress, however, that there is a
large literature on factored (multiagent) decision making that has been motivated by many different applications, such
as smart grids, warehouse commissioning, etc, and that Bayesian networks (the tool we use to represent the factored
structure) are one of the most applied AI techniques in history (even if perhaps overtaken by NNs by now).

When sufficient time is available for online planning, the benefits in the traffic domain are indeed moderate (**R1**).
However, in many real-world decision problems, the online decision time is limited, and precisely for those cases our
approach significantly outperforms the baselines (e.g., when <10s per action allowed in Fig. 4d). To further test this,
Figure 1 shows real-time planning results in the grab a chair domain, as suggested by **R3**. These further demonstrate
that the advantage of our approach in time-constrained settings, as the global model of the problem gets more complex.

**Further clarifications**: **R3**: learning models that matter with value, goal and task awareness is indeed relevant to the
idea of capturing the impact from the rest of the environment by predicting only the influence source variables. However,
these approaches do not perform explicit abstraction as ours does and therefore are complimentary. **R2**: we chose to
use POMCP in unmodified form in our experiments to make it easier to interpret the benefits of our approach, but
indeed our approach can be combined generally with other planners and more advanced particle filters, and therefore
all known improvements from SIR can be applied. **R4**: it is not possible to define a baseline local simulator with "no
influence" because by definition the influence destination variables have dependency on the source variables. We agree
that attention mechanisms are a promising avenue for learning influence representations (not the focus of this paper)
and thus are a promising direction of future work. As for the grab a chair domain (**R4**), we performed online planning
for a finite horizon of 10 (line 198-199), which explains the choice for undiscounted reward $\gamma = 1$. At every time step,
each agent can only decide to grab the chair on their left or right and therefore it is not possible for an agent to obtain
more than one chair at a time step (line 33, 388-389).

[Meta-Review · NeurIPS 2020]

The paper addresses a problem of computationally expensive forward simulation in POMDPs. The authors propose a POMDP solver that uses influence-based attention to cheaply estimate substates, reducing overall cost of the forward simulation. There is very little work in reducing the cost of forward simulation, which makes this work interesting and important. The method is well founded, with somewhat weak in the evaluations. The evaluations shows significant benefit in two domains, but omit in-depth analysis of the algorithm. In the light of the after the authors' responses and reviewers' discussion, the authors should include: - The discussion from their response on the differences to the exogenous and endogenous factors in POMDPs, and clearly explain that the influence-based attention is more general.